# Engineering atomic-scale magnetic fields by dysprosium single atom magnets

A. Singha [1,2,3,7✉], P. Willke [1,2,4,7], T. Bilgeri[5,7], X. Zhang[1,2], H. Brune [5], F. Donati[1,6], A. J. Heinrich [1,6✉] &
T. Choi [1,6✉]

Atomic scale engineering of magnetic fields is a key ingredient for miniaturizing quantum devices and precision control of quantum systems. This requires a unique combination of magnetic stability and spin-manipulation capabilities. Surface-supported single atom magnets offer such possibilities, where long temporal and thermal stability of the magnetic states can be achieved by maximizing the magnet/ic anisotropy energy (MAE) and by minimizing quantum tunnelling of the magnetization. Here, we show that dysprosium (Dy) atoms on magnesium oxide (MgO) have a giant MAE of 250 meV, currently the highest among all surface spins. Using a variety of scanning tunnelling microscopy (STM) techniques including single atom electron spin resonance (ESR), we confirm no spontaneous spin-switching in Dy over days at ≈ 1 K under low and even vanishing magnetic field. We utilize these robust Dy single atom magnets to engineer magnetic nanostructures, demonstrating unique control of magnetic fields with atomic scale tunability.

[1] Center for Quantum Nanoscience, Institute for Basic Science (IBS), Seoul, Republic of Korea. [2] Ewha Womans University, Seoul, Republic of Korea. [3] Max Planck Institute for Solid State Research, Stuttgart, Germany. [4] Physikalisches Institut, Karlsruhe Institute of Technology, Karlsruhe, Germany. [5] Institute of Physics, École Polytechnique Fédérale de Lausanne, Lausanne, Switzerland. [6] Department of Physics, Ewha Womans University, Seoul, Republic of Korea. [7] These authors contributed equally: A. Singha, P. Willke, T. Bilgeri. ✉email: a.singha@fkf.mpg.de; heinrich.andreas@qns.science; choi.taeyoung@qns.science

Single lanthanide atoms adsorbed on surfaces or incorporated in molecular complexes are being pursued in recent years for applications in quantum information technology and high-density magnetic data storage. Major advances in these domains include all-electrical read-out[1] and coherent control of their nuclear spins[2], the use of atomic clock transitions to protect their spins against dipolar decoherence[3], the discovery of single atom magnets[4,5], and the ability to manipulate their spin states[6]. Recently, a distinct class of molecular magnets containing a single Dy atom has even pushed the limit of operational temperatures up to 80 K by enhancing the MAE[7,8]. Higher uniaxial MAE acts as a barrier against spontaneous reversal of magnetic spins[9–11], thus enhancing their magnetic lifetimes. The charge balance in such molecular magnets, however, necessitates the use of anionic counterparts, posing limitations on isolating and supporting them on a solid-state substrate. In contrast, single lanthanide atoms can be directly adsorbed on insulating MgO substrates, and due to the linear bond formed at the oxygen site, large MAE values are also expected[12]. Among the late lanthanides, terbium (Tb), Dy, and holmium (Ho) atoms exhibit large MAE in a uniaxial crystal field (CF)[12,13], which led to the first few lanthanide-based single-chain magnets with slow magnetic relaxation[14,15]. In particular, Dy ions and diatomic units deposited on MgO are predicted to possess significantly larger MAE compared to their counterparts containing Ho[12]. Additionally, the seminal single-atom magnet, Ho on MgO[4], despite its long magnetic lifetime, lacks resilience against slow sweep rates of tip-magnetic fields[16].

In this work, we report the first single-atom magnet on a surface that maintains stability at zero external magnetic field as well as against slow magnetic field sweeps, and hence can be used to create atomically localized magnetic fields. We observe magnetic stability of several days in Dy atoms adsorbed on MgO at $\approx 1$ K with a giant MAE of 250 meV, using a low-temperature STM combined with single-atom ESR capability[17]. Using a spin-polarized tip (SP-tip) we measure random telegraph signal of the two magnetic states in Dy atom and reveal that the magnetization reversal becomes possible only via scattering with tunnelling electrons of energies higher than 140 meV, which is twice compared to the case of single Ho atoms adsorbed on MgO/Ag (100)[18]. By employing tip-field sweeps[19] on sensor Fe atoms[20], we directly measure the magnetic dipolar fields from Dy ($B_{Dy}$), which show absence of all major relaxation pathways at low and even zero external magnetic field ($B_{ext}$)[21], unlike Dy-based single-molecule magnets[7,22,23]. We harness this zero-field magnetic stability in single Dy atoms in several engineered Fe–Dy structures, to demonstrate atomically precise control of magnetic fields.

## Results

Figure 1 a shows a constant current STM image highlighting both Fe and Dy single atoms adsorbed atop an oxygen-site of a bilayer MgO patch grown on Ag(001) (see "Methods"). For characterizing the magnetic stability of the Dy atom, we first investigate it with an SP-tip prepared by transferring several Fe atoms from the MgO surface to the tip-apex[24]. SP-tips offer spin-sensitive read-out via tunnelling magnetoresistance. Thus, a magnetic contrast between two distinct Dy-spin orientations is recorded as switching events in the time-trace of the tip–sample distance ($\Delta z$) for fixed tunnelling currents (Fig. 1b)[6,25,26]. The switching rate increases with bias voltage $V_{dc}$, while no magnetization reversal is ever observed for $|V_{dc}| < 140$ meV. From the bias-voltage-dependent switching events measured at 5 T out-of-plane magnetic field (Fig. 1c), we identify two dominant thresholds, at 155 mV and 235 mV (Table S2), which are twice compared to the respective values reported for single Ho atoms on MgO/Ag

(001)[18]. In order to identify the magnetization reversal mechanisms associated with these thresholds, we perform point-charge-based multiplet simulations[27]. Figure 1d illustrates the resulting energy level distribution of a single Dy atom in $4f^9$ configuration, indicating an out-of-plane total MAE of 250-meV with a ground state Kramer doublet of $\langle J_z \rangle = \pm \frac{15}{2}$ and a magnetic moment of $9.9 \mu_B$. Note that the $4f^{10}$ configuration results in significantly reduced MAE (Fig. S8). Due to the fourfold symmetry of the O-adsorption site, the eigenstates of the system are linear combinations of several $J_z$ states[28] separated by $\Delta m_J = \pm 4$. The resulting state mixing is stronger for states with lower $\langle J_z \rangle$ (Fig. 1d). In presence of the tunnelling electrons, this opens up transitions between an initial and a final state with probabilities defined by the interaction operator $\boldsymbol{J} \cdot \boldsymbol{\sigma} = J_z \sigma_z + \frac{1}{2}(J_+ \sigma_- + J_- \sigma_+)$, where $J_{+,-}$ are the ladder operators of the Dy angular momentum $J$, and $\sigma$ is the spin operator of the tunnelling electrons[29]. The selection rule for $\boldsymbol{J} \cdot \boldsymbol{\sigma}$ allows transitions for $\Delta J_z = 0, \pm 1 (\text{mod} 4)$, where $\Delta J_z$ is the difference between initial and final state values of $\langle J_z \rangle$. Consequently, we identify seven major routes of different intensities within 146–248 meV, which enable spin–flip events (Fig. 1d). The corresponding calculated switching probabilities for different $\Delta m$ transitions are in good agreement with experimentally measured thresholds shown in Fig. 1c (see Supplementary section 8).

Although the giant MAE of 250 meV largely reduces the probability of spin-transitions over the anisotropy barrier, it does not ensure magnetic stability against quantum tunnelling. However, following Kramer's theorem[30], the ground state of Dy with half-integer value of $\langle J_z \rangle$ should remain doubly-degenerate at $B_{ext} = 0$ T, thereby implying a vanishing transition probability between $\langle J_z \rangle = \pm \frac{15}{2}$. In order to verify this zero-field magnetic stability of Dy, we employ a specific ESR detection method operating in zero external magnetic field and using sweeps of the tip magnetic field only ($B_{tip}$)[19]. We probe the stability of the Dy spin states via coupled sensor Fe atoms in engineered Fe–Dy pairs of varying interatomic distances $d$ (see Supplementary section 6). As illustrated in Fig. 2a, the radio frequency (RF) is kept constant ($f_0$) and tip-field sweeps are achieved by varying the tunnelling current at constant $V_{dc}$, thereby changing tip–sample distance (see "Methods" and Supplementary section 2). For the following, it is important to note that the SP-tips used in this work for measuring tip-field sweep ESR showed magnetic bistability in low fields ($|B_{ext}| < 60$ mT), with lifetimes typically shorter than the timescales of our measurements (see Supplementary section 2). This results in two different orientations of the tip-fields with respect to all static magnetic fields ($B_{static}$). Thus, considering only the out-of-plane projections from both $B_{tip}$ and $B_{static} = B_{Dy} + B_{ext}$, the generic condition for driving tip-field sweep ESR in Fe atoms in strong tip-field regimes ($|B_{tip}| > |B_{static}|$) can be written as (see "Methods"):

$$\frac{hf_0}{2\mu_{Fe}} = |\pm B_{tip} + B_{static}| = |B_{tip}| \pm |B_{Dy} + B_{ext}| \qquad (1)$$

Here, $h$ is Planck's constant and $\mu_{Fe}$ is the magnetic moment of the sensor Fe atom. In the absence of any static magnetic field ($B_{Dy} = 0$ and $B_{ext} = 0$), spin resonance occurs when the tip-field induced Zeeman splitting between the two lowest lying states of the Fe atom ($2\mu_{Fe}B_{tip}$)[17] matches $B_0 = hf_0$, leading to a single resonance (Fig. 2b). However, in the presence of a static dipolar field from Dy, ($B_{Dy} \neq 0$ and $B_{ext} = 0$), the resonance results from two different relative alignments of $B_{tip}$ and $B_{Dy}$. Consequently, the resonance condition is satisfied at two distinct tip-fields centred around $\frac{hf_0}{2\mu_{Fe}}$, i.e., $B_{tip}^1 = \frac{hf_0}{2\mu_{Fe}} - B_{Dy}$ and $B_{tip}^2 = \frac{hf_0}{2\mu_{Fe}} + B_{Dy}$. This interpretation is in excellent agreement with our observation of two ESR peaks for the Fe atom in a Fe–Dy pair at $B_{ext} = 0$ T

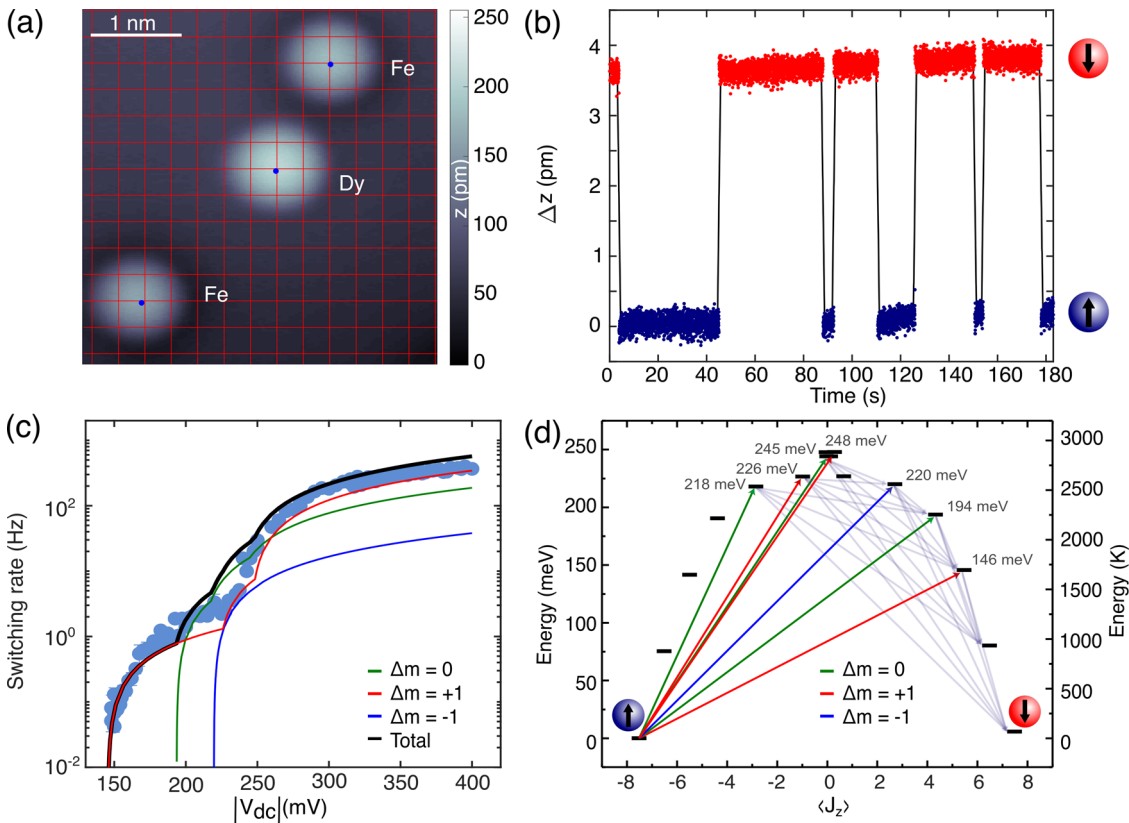

**Fig. 1 Magnetic contrast and large magnetic anisotropy in single Dy atoms on MgO. a** STM image of single Fe and Dy atoms adsorbed on a bilayer MgO patch grown on Ag(001). The intersections of the red lines mark the oxygen sublattice of the MgO surface ($T = 0.7$ K, $V_{dc} = 100$ mV, $I_t = 20$ pA). **b** Spin-polarized detection of time-dependent change in apparent height $\Delta z$ atop Dy atom exhibiting its two possible magnetic orientations ($T = 1.8$ K, $B_{ext} = 5$ T, $I = 1.5$ nA, $V_{dc} = -156$ mV). **c** Dy spin-switching rate $\tau$ as a function of bias voltage $V_{dc}$. For recording switching rates spanning 3–4 orders of magnitude, we acquired data at $I_t = 1.5$ nA (150–250 mV), $I_t = 0.15$ nA (250–320 mV) and $I_t = 0.015$ nA (315–400 mV) and multiplied them by 1, 10 and 100, respectively. Solid lines are calculated switching rates from different $\Delta m$ transitions using multiplet analysis ($T = 1.8$ K, at $B_{ext} = 5$ T). Error bars on each data point account for standard deviation and error propagation for fits to the residence times in both up and down state of Dy. **d** Level scheme inferred from multiplet calculations at $B_{ext} = 5$ T showing a giant MAE of 250 meV, ground state with $\langle J_z \rangle = \pm 15/2$, and seven routes within 146–248 meV for spin–flip transitions in Dy atoms. Bright colour-coded arrows with corresponding energies (in grey), represent transitions for different $\Delta m$, same as in (**c**). Following these transitions, a series of de-excitation processes (grey arrows) cause a complete spin reversal.

(Fig. 2b). Note that the separation between the two resonance peaks in Fe–Dy pairs directly provides a measure of the out-of-plane magnetic field generated by a single Dy atom ($B_{Dy}$). Within magnetic dipole approximations this is proportional to the out-of-plane magnetic moment of the Dy atom ($\mu_{Dy}$) as $B_{Dy} = \frac{\mu_0 \mu_{Dy}}{4\pi} \times d^{-3}$, where $\mu_0$ is the vacuum permeability.

The two-resonance feature also appears at small non-zero values of $B_{ext}$ for isolated Fe atoms as shown in Fig. 3a (left panel). This illustrates that the effect of the Dy dipolar field on a neighbouring Fe atom (Fig. 2b, top) is the same as having a static external magnetic field only. The positions of the two ESR peaks evolve linearly with $B_{ext}$ for all cases (Fig. 3a). However, for the Fe–Dy pairs the two ESR peaks merge when the external magnetic field compensates the dipolar field from the Dy atom ($B_{ext} = -B_{Dy}$), thereby shifting them with respect to the isolated Fe case (Fig. 3b). From these measurements we infer the dipolar field of the Dy atom for several Fe–Dy pairs of varying interatomic distance (Fig. 3c). Note that we observe an excellent agreement between frequency sweep and tip-field sweep ESR at two different set frequencies which suggests that tip-fields are not influenced by the presence of the Dy atom (see Supplementary section 2). We fit the $|B_{Dy}|$ values obtained from all ESR measurements to the explicit distance dependence $B_{Dy} = \frac{\mu_0 \mu_{Dy}}{4\pi} \times d^{-3}$ and obtain $\mu_{Dy} =$

$10.1 \pm 0.3 \mu_B$, in agreement with the multiplet analysis and the atomic value for Dy in gas phase.

Note that we never observe any spin-switching in Dy over days against repeated cycles of $B_{ext}$ ramps within $\pm 30$ mT (see Supplementary section 3), despite an isotopic composition which bears the possibility of spin–flip transitions at low magnetic fields via hyperfine interaction in $^{161}$Dy (19% natural abundance) and $^{163}$Dy (25% natural abundance). This is in contrast to the case of single Ho atoms on MgO, where spin-switching was observed at slow magnetic field sweeps[16]. Moreover, the magnetic state of the Dy atom is also stable against high magnetic fields of 5 T and heating of at least up to 15 K (Fig. S7). Altogether, these results lead us to conclude that Dy is a single atom magnet atop O-site of MgO, even in the limit of vanishing magnetic field. The observed magnetic stability is supported by our multiplet analysis which indicates that the g-factor is essentially uniaxial with negligible transverse component (see Supplementary section 8).

Finally, we demonstrate the versatility of Dy single-atom magnets by combining their long-term magnetic stability with the ability to control their spin state by high-energy tunnelling electrons. To illustrate this, we measure tip-field sweep ESR at $B_{ext} = 0$ T on the Fe sensor with increasing number of surrounding Dy atoms ($N$), as presented in Fig. 4a. These Fe–Dy$_N$ structures are

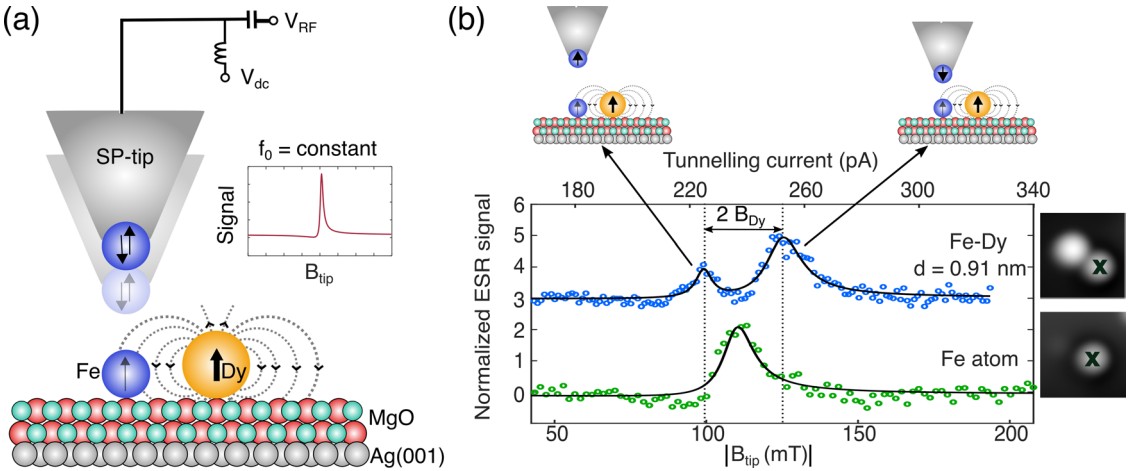

**Fig. 2 Zero-field magnetic stability of Dy atom. a** Schematic representation of our experimental setup for tip-field sweep ESR acquired on an Fe sensor atom at a fixed radio frequency $f_0$. Arrows with opposite orientations at the tip-apex represent the bistable nature of the out-of-plane component of the tip-field at vanishing external magnetic fields. Arrows on Fe and Dy atoms indicate respective out-of-plane magnetic moments. **b** Tip-field sweep ESR measured on an isolated Fe atom (green, bottom) and Fe atom in a Fe-Dy pair (blue, top) at $B_{ext} = 0$ T. Solid lines are fits to the data using a Fano–Lorentzian function (see "Methods"). Tip-field sweeps are achieved by varying the tunnelling current and thereby the tip–sample distance at a fixed dc bias ($T = 0.4$ K, $V_{dc} = -50$ mV, $V_{RF} = 27.5$ mV, $f_0 = 16.38$ GHz). The schematics highlight two opposite spin orientations of the paramagnetic tip. For a given spin-up configuration of the Dy atom, one of these tip-states satisfies the resonance condition at a lower tip–sample distance compared to the other. From the separation of the two resonance peaks we extract $B_{Dy} = 13.0 \pm 0.3$ mT. Insets show STM images of the single Fe atom and the Fe–Dy pair, where the cross indicates the position of the tip during the ESR scan and $d$ defines the distance between the Fe and Dy atoms (Image size: $2.5 \times 2.5$ nm$^2$, $T = 0.4$ K, $V_{dc} = 100$ mV, $I_t = 20$ pA).

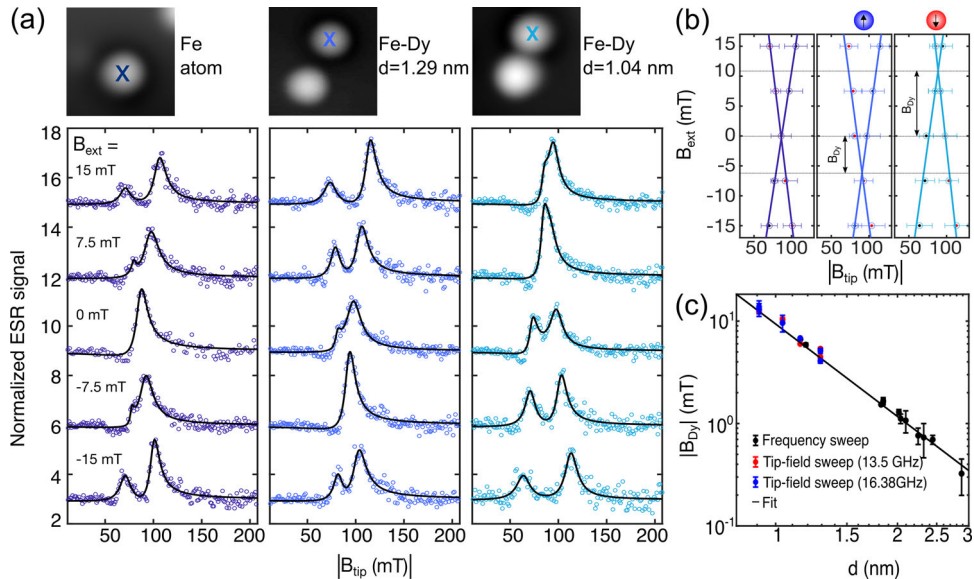

**Fig. 3 Measuring the stable dipolar magnetic field of a single Dy atom. a** $B_{ext}$ dependence of tip-field sweep ESR for an isolated Fe atom (left panel), and two Fe–Dy pairs of varying distances $d$ (in middle and right panel; $T = 0.5$ K, $f_0 = 13.5$ GHz, $V_{RF} = 27.5$ mV, $V_{dc} = -50$ mV). For the case of Fe–Dy pairs, the resonance condition is satisfied at two distinct tip-fields given by, $B_{tip}^1 = \frac{hf_0}{2\mu_{Fe}} - (B_{ext} + B_{Dy})$ and $B_{tip}^2 = \frac{hf_0}{2\mu_{Fe}} + (B_{ext} + B_{Dy})$. Solid lines are fits to the data (see "Methods"). The colour-coded cross marks on the corresponding STM topographies indicate the position of the sensor Fe atom where the ESR measurements were taken (Image size: $3 \times 3$ nm$^2$, $V_{dc} = 100$ mV, $I_t = 20$ pA, $T = 0.5$ K). **b** ESR peak positions as a function of tip-field as extracted from the fits for all three datasets shown in (**a**). Error bars indicate standard deviation from fitting with equation (3). The intersection of the straight line fits provide $B_{Dy} = -5.4 \pm 0.1$ mT (middle) and $10.4 \pm 0.4$ mT (right), where the sign indicates two opposite Dy magnetic orientations, as in the schematics. **c** The dipolar magnetic field of single Dy atom $B_{Dy}$ as a function of $d$ in different Fe–Dy pairs from both tip-field sweep at two different set frequencies (red and blue) and frequency sweep (black) ESR. For tip-field sweep data, error bars indicate propagated total errors from linear fits shown in (**b**). For frequency sweep data, error bars indicate propagated total errors from fitting frequency sweep ESR spectra such as in Fig. S5. The solid line is a fit to the data following magnetic dipole approximation, $B_{Dy} = \frac{\mu_0\mu_{Dy}}{4\pi} \times d^{-3}$, resulting in $\mu_{Dy} = 10.1 \pm 0.3\mu_B$.

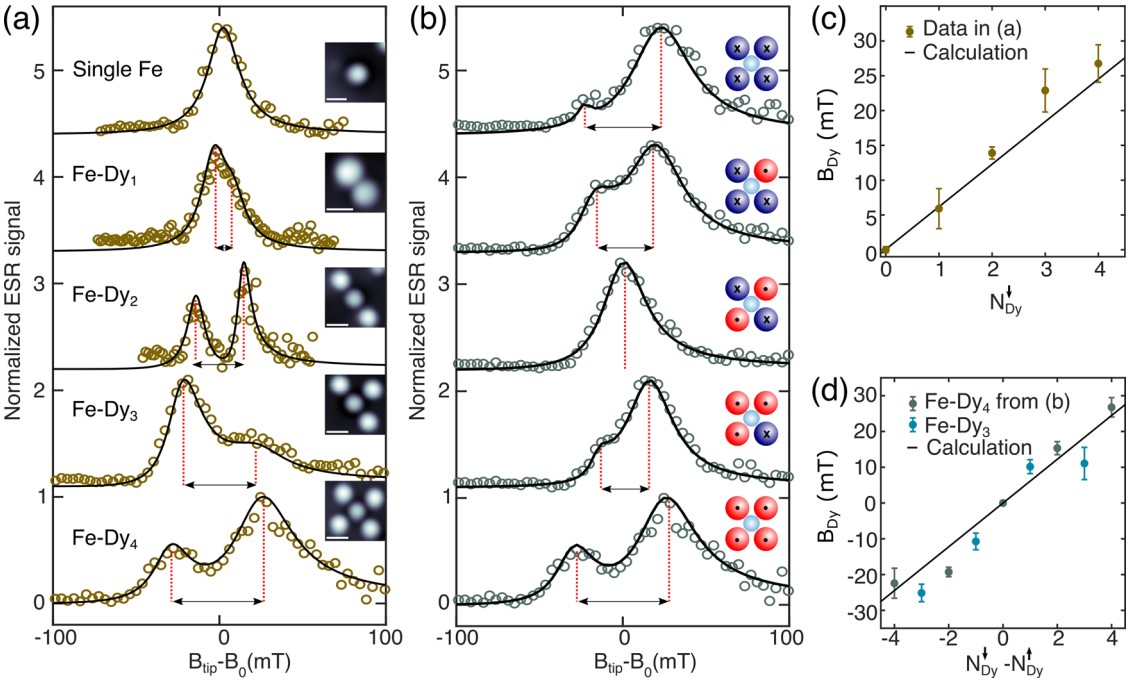

**Fig. 4 Atomic engineering of local magnetic fields. a** With increasing number of surrounding Dy atoms, tip-field sweep ESR at $B_{ext} = 0$ T demonstrates the increase of $B_{Dy}$ experienced by the sensor Fe atom. Dy atoms are sequentially placed 1.15 nm apart from the central Fe atom where ESR spectra are recorded. Solid lines are fits to the data (see "Methods"). (Fe-Dy$_{N=0,1,3,4}$: $T \approx 1.3$ K, $f_0 = 21.4$ GHz, $V_{RF} = 15$ mV, $V_{dc} = -50$ mV; Fe-Dy$_2$: $T = 0.8$ K, $f_0 = 16.25$ GHz, $V_{RF} = 10$ mV, $V_{dc} = -50$ mV). Insets show corresponding STM images. (Scale bar: 1 nm; Fe-Dy$_{N=0,1,2,3}$: $V_{dc} = -100$ mV, $I_t = 20$ pA, $T \approx 1.3$ K; Fe-Dy$_4$: $V_{dc} = 80$ mV, $I_t = 20$ pA, $T \approx 1.3$ K). **b** Modifying $B_{Dy}$ by selectively manipulating the magnetic states of individual Dy atoms in Fe-Dy$_4$ ($T \approx 1.3$ K, $f_0 = 21.4$ GHz, $V_{RF} = 15$ mV, $V_{dc} = -50$ mV). Insets show Fe-Dy$_4$ spin schematics (red spheres: Dy spin down configuration (cross); blue spheres: spin up (dot) configuration; cyan spheres: Fe spin). For both (**a**) and (**b**), $B_O$ corresponds to the tip-field required for driving ESR in a reference Fe atom. Solid lines are fits to the data (see "Methods"). **c, d** The dipolar magnetic field of Dy measured in different Fe-Dy$_N$ structures shown in (**a**) and (**b**), respectively. Additional data from an Fe-Dy$_3$ is shown in (**d**). The solid lines in (**c**) and (**d**) indicate calculated values using $B_{Dy} = \frac{\mu_0 \mu_{Dy}}{4\pi} \times d^{-3}$. Error bars in (**c**) and (**d**) indicate propagated total errors from fitting ESR spectra shown in (**a**) and (**b**), respectively.

built by positioning one Dy atom at a time at fixed Fe–Dy distance of 1.15 nm (Fig. S6). Subsequently, we prepare their magnetic states into spin down configurations in a site selective manner by injecting high-energy tunnelling electrons ($|V_{dc}| > 150$ mV). Thus the total magnetic dipolar field on the sensor Fe atom increases in structures with higher number of Dy atoms, resulting in larger separations between the two resonance peaks (Fig. 4a). Furthermore, selective manipulations of individual Dy spins in Fe–Dy$_4$ also allow us to modify the total magnetic field on the sensor Fe atom (Fig. 4b). Consequently, the separation between the two ESR peaks varies within $\approx \pm 25$ mT (Fig. 4d). As illustrated in Fig. 4c, d, both approaches lead to a linear increase of the total magnetic dipolar field with the number of Dy atoms, which can be additionally tuned by their relative spin-orientation, thus allowing unique local controls of atomic scale magnetic fields. Such deterministic and local control of magnetic fields can be highly desirable as atomic gates for nanoscale logic devices[31], as well as for surface-based quantum architectures. In addition, interacting Dy spin-centres within a surface-based quantum network can exhibit collective magnetic behaviour with high blocking temperature and slow magnetic relaxation, similar to lanthanide-based single-chain magnets[14,15].

## Methods

**Sample preparation.** The Ag(001) surface was prepared by several cycles of sputtering and annealing. For subsequent MgO growth, the crystal was heated to 700 K and exposed to Mg from a crucible evaporator in an oxygen partial pressure of $1 \times 10^{-6}$ mbar. Under these conditions, 40 min exposure yields an average of two monolayers of MgO. Next, the sample was cooled down to room temperature within 15 min, and was transferred to the cold STM (4 K). Prior to single atom deposition, the sample manipulator was pre-cooled by touching the sample for

20 min. Subsequently, the sample was quickly taken into an exchange chamber, where depositions of Fe and Dy were performed within a few seconds on to the cold sample using an electron beam evaporator and at a base pressure of $<7 \times 10^{-10}$ mbar. For STM measurements we used a PtIr tip, with a tip apex presumably silver-coated due to repeated indentation into the silver substrate for tip preparation. Tunnelling bias voltages were applied to the tip, however, the $V_{dc}$ values are expressed with an additional negative sign, i.e., with respect to the sample, as it is conventionally done.

**ESR measurements.** For the ESR measurements[17], an RF generator (Keysight E8257D) was used to generate the radio frequency signal, which was subsequently added to the DC bias voltage using a bias-tee (SigaTek SB15D2) located outside of the vacuum chamber. A lock-in amplifier (Stanford Research Systems SR860) operating with on-off modulation at 95 Hz was used for signal detection. For conducting the tip-field sweep ESR, the RF-generator was fixed at a constant frequency and output power. Next, the STM feedback loop was kept engaged with low feedback gain and the tunnel current set point was swept. Consequently, the tip–sample distance was swept while the lockin-signal was monitored.

Note that the tip-field sweep ESR measurements are typically performed in strong tip-field regimes satisfying $|B_{tip}| > |B_{Dy} + B_{ext}|$. Thus, for Fe atoms in Fe–Dy pairs, the generic resonance condition can be written as

$$\frac{hf_0}{2\mu_{Fe}} = \begin{cases} |B_{tip}| + |B_{Dy} + B_{ext}|, & \text{if } B_{tip}(B_{Dy} + B_{ext}) > 0 \\ |B_{tip}| - |B_{Dy} + B_{ext}|, & \text{if } B_{tip}(B_{Dy} + B_{ext}) < 0 \end{cases} \quad (2)$$

Given its stable magnetic orientation, the Dy atom induces a constant dipolar magnetic field on the Fe sensor. This results in two resonance peaks for all values of $B_{ext}$ used in this work, except for the merging points, i.e., $B_{ext} = -B_{Dy}$. In contrast to the case of Dy atoms, Fe atoms on MgO exhibit significantly shorter magnetic lifetimes[24]. Following Eq. (2), this gives rise to a maximum of four ESR peaks for Fe atoms in Fe–Fe pairs as shown in Supplementary section 4.

**Fitting ESR peaks.** In order to determine the position of the resonance during the tip-field sweeps, we fit the experimental data shown in Fig. 2b, 3a, and 4a, b to a

Fano–Lorentzian of the form

$$\Delta I = I_{\text{peak}} \times \frac{1}{q^2 + 1} \times \frac{(1 + \delta \times q)^2}{1 + \delta^2} \tag{3}$$

Here, $I_{\text{peak}}$ is the amplitude at the resonant tunnelling current and $q$ is the Fano factor, arising from an additional homodyne detection of ESR[32]. In addition, $\delta = \frac{B_{\text{tip}} - B_0}{\tau/2}$, where $\tau$ is the linewidth of the peak and $B_0$ is the tip magnetic field for which the resonance occurs in an isolated Fe atom. The ESR signal was always normalized to the set point tunnel current for comparison. Finally for all datasets a polynomial background of degree 2 was subtracted, since the signal intensity increases with tunnel current, due to an increase of the number of read-out tunnel current electrons.

## Data availability

The datasets generated during and/or analysed during the current study are available from A.S. (a.singha@fkf.mpg.de) upon reasonable request.

## Code availability

There is no mathematical algorithm or custom code that is deemed central to the conclusion of this manuscript. However, the custom codes that were simply used for data analysis are available from A.S. (a.singha@fkf.mpg.de) upon reasonable request.

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

## Acknowledgements

All authors acknowledge the Institute for Basic Science for grant IBS-R027-D1. P.W. acknowledges funding from the Emmy Noether Programme of the DFG (WI5486/1-1). T.B. acknowledges support from the Swiss National Science Foundation under Project no. 200020_176932.

## Author contributions

A.S. conceived the idea. A.S., P.W., T.B., and X.Z. performed the experiments. A.S., P.W., and T.B. analysed the data. F.D. performed the multiplet analysis. T.C., F.D., H.B., and A. J.H. supervised the project. A.S. wrote the manuscript with contributions from all co-authors. All authors discussed the results.

## Funding

## Competing interests

The authors declare no competing interests.
