## [Peer Review File · Nature Communications]

Reviewers' Comments:

Reviewer #1:

Remarks to the Author:

This paper reports on magnetic stability of a single Dy atom adsorbed on MnO layer. By using ESR-STM the authors successfully demonstrated the stability and precisely measured magnetic fields induced by the adsorbed atom. These subjects are quite interesting and important in the community of nano- and atomic-scale magnetism, and nanotechnology in general. The presented techniques are quite challenging, and the presented experimental results and interpretation are superb and convincing indeed. I thus fully support for the publication of this paper in the journal. The following is a list of minor comments that may help improve it.

1. It seems strange to me why the tip magnetization always exhibits bi-stability. In the supplemental, the authors present the case where the switching rate is slow and thus only one peak appears in the tip-sweep ESR. In both cases, however, the tips are bi-stable. A chunk of 30 Fe atoms may have anisotropy; it could be easy-axis or hard-axis. The magnetization does not always have to be bi-stable, I think. I wonder why the results always show the bi-stability, and I would like to know the authors' comment on it.
2. In the case of weak anisotropy, the tip magnetization may be modified (tilted) due to the presence of magnetic field induced by Dy. If tilted during approaching, out-of-field component changes with the distance, and this might cause systematic error in the estimation of the magnetic field the sensor Fe atom feels. I would like to know such modification is really negligible or not.

Reviewer #2:

Remarks to the Author:

The manuscript "Engineering atomic-scale magnetic fields by dysprosium single atom magnets" details the investigation of dysprosium "single atom magnets", i.e. single slow-relaxing dysprosium magnetic centres. The technique used is rather new and the group is the undisputed leader in the area.

Using STM, the authors show that Dy atoms on MgO have a giant magnetic anisotropy energy, and reach a record value for all known surface spins. Moreover the magnetization of the Dy stays fixed for days (albeit at ≈ 1 K) even at zero field. The authors go on to demonstrate unique control of magnetic nanostructures at the atomic scale.

The paper is well written, structured in a logical manner and the data are solid and well presented. The main data treatment is very solid.

The paper has one point that is very obscure, though, and which would need more work or discussion. At zero magnetic field (or close to zero field) one would expect quantum tunneling of the magnetization. For a half-integer spin system, there is no way in which the zero-field degeneracy can be lifted. The reason for this is fundamentally rooted in time-reversal symmetry (Kramers's theorem) and spurious effects could not be responsible for suppressing the degeneracy. Similar effects should also be observed at level crossing.

The authors observe no such effect, and they do state that this is contrary to many observations up to date. I think anyway that a much stronger discussion, and possibly more datasets, would be in order, so as to provide a rationale for these observations and possibly to give an idea of the field dependence of the relaxation times etc...

Other than that, I think the paper could also benefit from considering that interacting Dy centres could give rise to structures akin to so-called single-chain-magnets, where Dy indeed played an important role in some of the early observations.

Otherwise, I think that the paper is very interesting and should be published in NComm.

1 Reply to Reviewer 1

We thank the reviewer for appreciating the high quality of our experimental data and interpretation, for supporting the publication of our work in Nature communications, and for recognizing it as "interesting and important in the community of nano-and atomic-scale magnetism". In what follows, we reply to the specific minor concerns raised by the reviewer.

- It seems strange to me why the tip magnetization always exhibits bi-stability. In the supplemental, the authors present the case where the switching rate is slow and thus only one peak appears in the tip-sweep ESR. In both cases, however, the tips are bi-stable. A chunk of 30 Fe atoms may have anisotropy; it could be easy-axis or hard-axis. The magnetization does not always have to be bi-stable, I think. I wonder why the results always show the bi-stability, and I would like to know the authors' comment on it.

As the referee has pointed out, the magnetic tips used for the measurements of zero-field ESR always exhibit magnetic bi-stability, albeit with varying lifetimes. We consider the cluster of Fe atoms adsorbed onto the tip apex as a single macrospin, resulting from the strong ferromagnetic coupling between the individual Fe atoms. The overall anisotropy energy of that cluster can conservatively be estimated by 30 times the atomic value for bulk Fe of $< 2.6 \mu\text{eV}$ (Nature **117**, pp. 753–754 (1926), Phys. Rev. B **57**, 9557 (1998)) to yield 0.078 meV, or as upper bound by taking 30 times the atomic value found for small Fe clusters on Al_2O_3 of $500 \mu\text{eV}$ (Phys. Rev. B **82**, 094409 (2010)) yielding 15 meV. In the first case, the tip magnetization switches more than once per second at $T > 0.045 \text{ K}$ and in the second at $T > 8 \text{ K}$. Magnetic bi-stability has been reported for few-atom clusters of Fe on both metal and insulating substrates (Science **335**, 6065, pp. 196-199 (2012); Science **339**, 6115, pp 55-59 (2013); Nature Communications **7**, 10454 (2016)). These configurations, though established on a sample surface, mimic the constellation on the tip pretty well, since the Fe atoms also form a cluster on a metal tip. We want to highlight that such a magnetic bi-stability is not expected for SP-tips possessing out-of-plane hard axis. This is due to the absence of doubly degenerate magnetic levels in such tips which would result in an almost zero net-magnetization. Moreover, we note that such tips would not be able to efficiently drive ESR transitions in the Fe atom due to the absence of the longitudinal component of the tip-field (Science Advances **6**, 40, eabc5511 (2020)). We also note that the magnetic bi-stability is lost for measurements at significantly higher fields (typically $\geq 60 \text{ mT}$) where one of the two magnetic states of the SP-tip is more favoured than the other, as already mentioned in the main text. Based on this, we can also estimate the magnetic anisotropy energy of the SP-tip as $30 * 3\mu_B * 60 \text{ mT} = 0.31 \text{ meV}$, which is well between the two conservative estimates presented above.

In order to improve clarity of this point, we have appended this discussion in the relevant section of the supplementary information on page 3-4.

- In the case of weak anisotropy, the tip magnetization may be modified (tilted) due to the presence of magnetic field induced by Dy. If tilted during approaching, out-of-field component changes with the distance, and this might cause systematic error in the estimation of the magnetic field the sensor Fe atom feels. I would like to know such modification is really negligible or not.

We thank the reviewer for drawing our attention to this point. We note that such effects are negligible compared to other sources of uncertainties in our measurements at vanishing magnetic fields. Firstly, this is supported by the perfect agreement between tip-field sweep and frequency sweep ESR, as shown in Fig. 3c of the main text. If the Dy atom continues to modify the out-of-plane component of the tip-magnetization during a tip-field sweep ESR measurement, systematic errors would occur in the estimation of the magnetic field sensed by the sensor Fe atom, as appropriately pointed out by the reviewer. This stems from the fact that for closer Fe-Dy distances the Dy field increases. However, this would also result in a strong deviation between the data obtained from tip-field sweeps and those gathered from frequency sweeps where the larger tip-sample distance leaves the tip-magnetization fully unaffected. In contrast, our data shown in Fig. 3c exhibit an excellent agreement between tip-field sweep and frequency sweep ESR carried out with different SP-tips for several Fe-Dy pairs of varying interatomic distances. Moreover, we also find an excellent agreement between tip-field sweep measurements obtained at two different fixed frequencies ($f_0 = 13.5$ GHz and 16.38 GHz). Note that Fig. 3c in the main text has now been updated by including the measurements at $f_0 = 16.38$ GHz. Changes in the set frequency would necessitate measurements at different range of tip-fields which implies different tip-sample distances. Any spurious effects or systematic error would thus result in disagreements between measurements taken at different values of f_0 , which is contrary to our observations. Furthermore, we also note that the effect of such systematic errors, if any, would also be visible from the B_{Dy} measurements shown in Fig. 4, where both the number as well as the orientation of different Dy atoms were modified for various structures. Consequently, if the Dy field would affect the tip field significantly, it would affect the determined Dy field, leading to a non-linear behaviour in Fig. 4c,d. In contrary, all measurements are in reasonable agreement with the magnetic moment of Dy obtained through two independent routes, *i.e.*, the measurements shown in Fig. 3c and our multiplet analysis, as shown from the respective linear trends.

Given these experimental evidences, we conclude that such systematic errors are negligible for our measurements, which is otherwise dominated by several other sources of errors such as instabilities in the tunnel junctions, uncertainties due to measurements with different SP-tips etc. In order to improve the clarity of the manuscript, we have included this discussion briefly within the main text (page 6) and in more detail in the supplementary page 5-6.

2 Reply to Reviewer 2

We thank the reviewer for her/his overall positive assessment of our work and for supporting its publication in Nature communications. In the following we address the specific concerns raised by the reviewer.

- The paper has one point that is very obscure, though, and which would need more work or discussion. At zero magnetic field (or close to zero field) one would expect quantum tunneling of the magnetization. For a half-integer spin system, there is no way in which the zero-field degeneracy can be lifted. The reason for this is fundamentally rooted in time-reversal symmetry (Kramers's theorem) and spurious effects could not be responsible for suppressing the degeneracy. Similar effects should also be observed at level crossing. The authors observe no such effect, and they do state that this is contrary to many observations up to date. I think anyway that a much stronger discussion, and possibly more datasets, would be in order, so as to provide a rationale for these observations and possibly to give an idea of the field dependence of the relaxation times etc...

We thank the reviewer for raising this point. We note that all attempts to see switching in the Dy atoms while sweeping the magnetic field across zero have failed. This includes measurements at vanishing magnetic fields for 10 different FeDy pairs including those shown in Fig. 3a, the four structures shown in Fig. 4a, as well as the FeDy₄ and FeDy₃ shown in Fig. 4b and 4d. Nearly 100 sweeps on each of these structures within ± 30 mT never exhibited any switching of the Dy magnetic orientation.

Firstly, Dy atoms on MgO are protected against quantum tunnelling events as the total ΔJ of 15 required for such transition is not an integer multiple of four, ruling out QTM within the four-fold symmetric adsorption site of the MgO surface. On a more general note, Dy atoms being Kramer's ions with a half-integer spin as the ground state $\langle J_z \rangle = \pm \frac{15}{2}$, are protected against quantum tunnelling of magnetization due to time reversal symmetry, as also noted by the reviewer. Due to this, the tunnel splitting between the doubly-degenerate levels should be exactly zero. In principle

this can be estimated from our multiplet analysis. However, in practice such estimations are less reliable due to the presence of several transverse terms and perturbations such as the dipolar field from the neighbouring spins as well as non-axial components of the tip-field, if any. Therefore, to verify the nature of the doubly-degenerate levels in the Dy atoms, we rely on the calculated values of transverse g-factors, as it is also commonly done in the molecular magnet literature (Inorg. Chem. **55**, 20, pp. 10043–10056 (2016); Science **362**, 6421, pp 1400-1403 (2018)). Our multiplet analysis shows that the doublet is essentially axial with $g_z = 19.7919$ and $g_x = 1.9 * 10^{-6}$. This value of g_x compares very well with the value reported by Guo et al. (Science **362**, 6421, pp 1400-1403 (2018)), where they quote $g_x = 0$ up to the 5th decimal digit (table S11 of the SI). Similar to this case, the Dy atom should also be very insensitive to any transverse field, so quantum tunnelling should be essentially suppressed. In order to improve the clarity concerning this point, we have added one sentence in the main text page 7 and added a detailed discussion including calculated values of g_x and g_z in the supplementary information page 17.

- Other than that, I think the paper could also benefit from considering that interacting Dy centres could give rise to structures akin to so-called single-chain-magnets, where Dy indeed played an important role in some of the early observations.

We are thankful to the reviewer for drawing our attention to the relevant earlier works on Dy single chain magnets. We have mentioned this through Ref. 14 and 15 in the introduction (page 2) and in the outlook (page 8) of the main text.

Reviewers' Comments:

Reviewer #1:

Remarks to the Author:

The authors replied to reviewers' comments appropriately. I do not have any further comments on the revised manuscript.

Reviewer #2:

Remarks to the Author:

The revisited version of "Engineering atomic-scale magnetic fields by dysprosium single atom magnets" by Aparajita Singha et al. provides a good revision of the manuscript, following all indications by both referees. I recommend its publication in its present form without further delay.

We thank you for considering our work for publication in Nature Communications following the overall positive assessments from both reviewers.

With kind regards,
Aparajita Singha (on behalf of the co-authors)

1 Reply to Reviewer 1

The authors replied to reviewers' comments appropriately. I do not have any further comments on the revised manuscript.

We thank the reviewer for appreciating the corrections made to the manuscript and for her/his overall positive assessment of our work.

2 Reply to Reviewer 2

The revisited version of "Engineering atomic-scale magnetic fields by dysprosium single atom magnets" by Aparajita Singha et al. provides a good revision of the manuscript, following all indications by both referees. I recommend its publication in its present form without further delay.

We thank the reviewer for her/his overall positive assessment of our work and for supporting its swift publication in Nature communications.